# TACLer: Tailored Curriculum Reinforcement Learning for Efficient Reasoning

## Abstract

Large Language Models (LLMs) have recently demonstrated remarkable performance on complex reasoning tasks, especially when equipped with long chain-of-thought (CoT) reasoning. However, eliciting long CoT reasoning typically requires large-scale reinforcement learning (RL) training, while often leading to overthinking with redundant reasoning steps. To improve learning and reasoning efficiency, while preserving or even enhancing performance, we propose TACLer, a tailored curriculum reinforcement learning framework that gradually increases the complexity of the data based on the model's proficiency in multi-stage RL training. Our framework features two core components: (i) tailored curriculum learning that determines what knowledge the model lacks and needs to learn in progressive stages; (ii) a hybrid *Thinking*/*NoThinking* reasoning paradigm that balances accuracy and efficiency by enabling or disabling the *Thinking* mode. Our experiments show that TACLer yields a twofold advantage in learning and reasoning: (i) it reduces computing effort by cutting training compute by over 50% compared to long thinking models and by reducing inference token usage by over 42% relative to the base model; and (ii) it improves accuracy by over 9% on the base model, consistently outperforming state-of-the-art *Nothinking* and *Thinking* baselines across four math datasets with complex problems.[1]

## 1 Introduction

Many recent developments in Large Language Models (LLMs) have focused on improving their ability to solve problems involving complex reasoning, which has long been considered one of the most challenging tasks (Wei et al., 2022; Wang et al., 2023; Zhou et al., 2023; Yue et al., 2024). Particularly, DeepSeek-R1 (Guo et al., 2025) and OpenAI's o1 (OpenAI, 2025) show that long chain-of-thought (CoT, Wei et al., 2022) sequences can be used to boost reasoning capabilities through large-scale reinforcement learning (RL), playing a crucial role in solving complex mathematical problems. The long CoT responses often contain a thinking process with reflection, backtracking, and self-validation, which helps to form the solution and arrive at the final answer. This test-time scaling paradigm (Wu et al., 2024; Muennighoff et al., 2025), enhances reasoning abilities at the cost of significantly increased token usage. Consequently, this introduces two major bottlenecks: (i) training models with reinforcement learning over large contexts demands substantial computational resources, e.g., a small model with 1.5 billion parameters requires 70,000 A100 GPU hours (Luo et al., 2025b); and (ii) so-called "overthinking" can occur during response generation, leading to redundant reasoning steps and an even heavier computational cost (Chen et al., 2025b; Luo et al., 2025a; Arora & Zanette, 2025).

To improve training efficiency, recent studies have proposed approaches such as the iterative lengthening scheme (Luo et al., 2025b) and curriculum learning (Song et al., 2025). Regarding overthinking, recent works explore efficient reasoning by reducing token usage in the thinking process, applying techniques such as such as length-based rewards (Arora & Zanette, 2025; Shen et al., 2025), thinking pruning (Luo et al., 2025a), and training models with adaptive thinking (Zhang et al., 2025; Tu et al., 2025). Nevertheless, these approaches often entail substantial additional training costs while only achieving limited reductions in response length, or may even degrade overall performance. Furthermore, models that rely on adaptive reasoning strategies tend to constrain users'

---

[1]Code and model will be made available upon publication.

options since in some scenarios users might prefer to access long thinking traces even for relatively simple tasks, for purposes such as interpretability, verification, and education. We argue that a more effective paradigm should enable the coexistence of *Thinking* and *Nothinking* modes within a model, balancing computational efficiency and model performance while providing flexible user control.

To this end, we propose `TACLer`, a tailored curriculum reinforcement learning framework, aimed at improving both learning and reasoning efficiency. First, unlike conventional curriculum learning that treats difficulty as a standardised concept, `TACLer` adapts the learning process to the model's evolving proficiency. Specifically, we gradually increase the complexity of the data based on the model's pass rate, enabling efficient learning of challenging tasks that are difficult to learn from scratch. Second, we adopt a hybrid *Thinking/NoThinking* reasoning paradigm that balances accuracy and efficiency through our curriculum learning paradigm. Our approach allows switching between concise reasoning for higher efficiency and long thinking reasoning for improved accuracy, thus reducing unnecessary computation at the user's discretion.

We demonstrate that `TACLer` reduces the required training compute by over 50% compared to other long thinking models such as DeepScaleR-1.5B-Preview (Luo et al., 2025b). Particularly, our extensive experimental results on four mathematical reasoning benchmarks (MATH 500, AMC, AIME 2024 and AIME 2025) show that `TACLer` achieves the best performance compared to various *Nothinking* and *Thinking* baselines, while significantly reducing the average number of token usage by over 42%, thereby effectively mitigating the overthinking problem.

In summary, the key contributions of our paper are as follows: (1) we introduce tailored curriculum learning, which determines what knowledge the model specifically lacks and needs to learn in progressive stages; (2) we incorporate both *Thinking* and *NoThinking* modes, showing that this can effectively balance reasoning accuracy and efficiency; (3) we propose `TACLer`, a novel RL-based framework that enables language models to learn more efficiently through tailored curriculum learning and to reason effectively via hybrid reasoning modes; and (4) we report extensive experiments and analysis to validate the efficiency of `TACLer` on both learning and reasoning.

## 2   PRELIMINARIES: LEARNING AND REASONING EFFICIENCY

The computational burden of training language models is often enormous, especially for tasks involving long thinking reasoning that require large context windows. For instance, Luo et al. (2025b) indicates that replicating DeepSeek-R1 experiments with $\geq$32K context and $\sim$8000 training steps would take at least 70,000 A100 GPU hours, even for a small model with 1.5B parameters. To mitigate this problem, Luo et al. (2025b) introduce an iterative lengthening scheme, which progressively increases the context length from 8K to 24K, thereby guiding the model to utilise the context more efficiently. However, as shown in Figure 1, more than 40% of the responses generated in the first training stage are still truncated, revealing a substantial inefficiency in the training process. Although Song et al. (2025) propose a curriculum learning framework that gradually increases the context length based on input length with the aim of facilitating progressive learning from easy to hard, almost 35% of responses re-

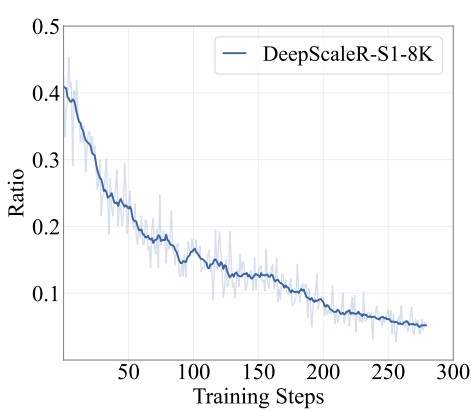

Figure 1: Response clipping ratio in the first training stage of DeepScaleR-1.5B-Preview.

main truncated in the first stage. The primary reason is that this is an arbitrary method for curriculum learning that does not take into account the model's mastery of specific problems.

Regarding reasoning efficiency, Figure 2 presents the comparison of DeepSeek-R1 and DeepScaleR in the *NoThinking* mode (see Sec. 3.2 for details), where the latter is trained based on the former to enhance the long CoT reasoning capability in the *Thinking* mode. Interestingly, DeepScaleR in the *NoThinking* mode exhibits consistent performance gains across questions of varying difficulty, accompanied by shorter responses. This shows that *Thinking* and *NoThinking* modes can coexist

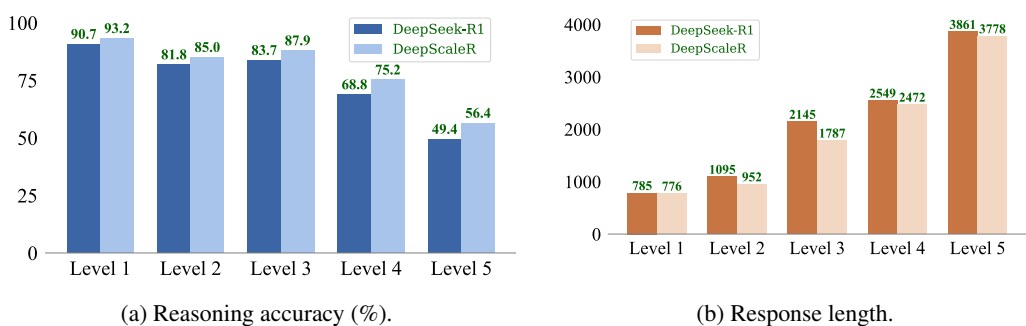

(a) Reasoning accuracy (%).    (b) Response length.

Figure 2: Comparison of DeepSeek-R1-Distill-Qwen-1.5B and DeepScaleR-1.5B-Preview using NoThinking mode across different difficulty levels of MATH500 dataset.

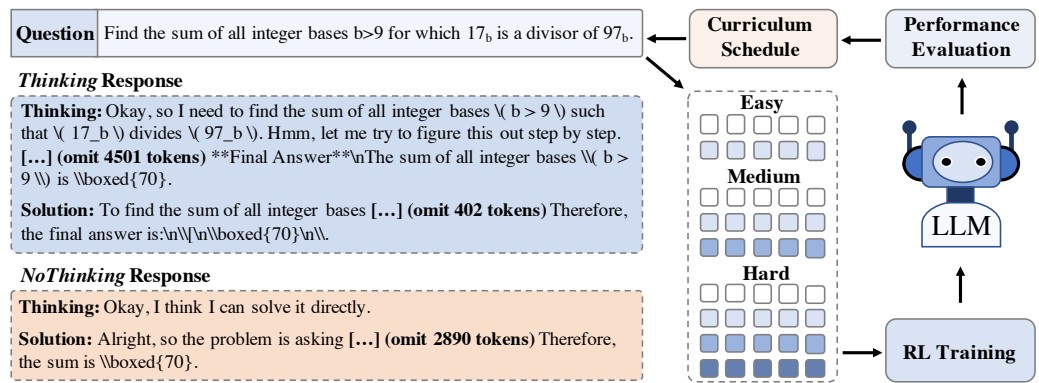

Figure 3: Overview of `TACLer`, our tailored curriculum reinforcement learning framework.

in the model and can be enhanced through RL training. Based on the above observation, a natural question arises: Can LLMs learn efficiently through a better curriculum learning strategy while leveraging a hybrid mode for efficient reasoning? To answer this question, we introduce `TACLer`, a novel RL-based framework that targets math reasoning tasks, which we detail in the next section.

## 3  METHODOLOGY

Figure 3 gives an overview of `TACLer` and its three components: (1) performance evaluation and curriculum schedule; (2) hybrid reasoning mode; and (3) RL training. Measuring performance is needed to create the curriculum schedule, with the goal of identifying the model's knowledge gap and enabling efficient learning across progressive RL stages. The hybrid reasoning mode, consisting of *Thinking* and *NoThinking*, achieves more efficient reasoning by balancing performance and computational cost during inference.

### 3.1  TAILORED CURRICULUM LEARNING

As shown in Section 2, the imbalanced data distribution poses a significant challenge: in the early training stages, a large portion of samples are excessively difficult. The resulting truncation not only leads to wasted computation but also hinders effective learning, thus substantially reducing training efficiency. To address this issue, we employ curriculum learning (Bengio et al., 2009), which gradually increases the complexity of the data samples throughout the learning process.

Our method leverages the notion of training data difficulty, where difficulty is defined by the model's ability to solve the corresponding problems. To obtain this assessment, we perform inference for all training data with greedy decoding and a context size of 8k, and categorise instances based on the responses and answers: (1) instances with the correct final answer; (2) instances with a complete

solution but an incorrect final answer; and (3) instances where the model fails to generate a complete solution. Based on these results, we can quantify problem-specific difficulty in terms of the model's proficiency. In turn, this enables us to organise training samples according to a tailored curriculum learning strategy, thereby allowing each model to tackle problems progressively from those it finds easier to solve to the harder ones.

During training, we merge samples from the first and second groups in the above inference results, ensuring that the training data is neither dominated by overly difficult nor overly simple problems, to implement a combined *review and learning* paradigm. This strategy enables the model to reinforce previously acquired knowledge while learning new skills, thus mitigating forgetting and enhancing overall learning efficiency. This entire process—inference, difficulty categorisation, curriculum learning—is repeated twice in our framework. In the third stage, we train on the full dataset using the model updated through the first two iterations to consolidate the learning results.

### 3.2 HYBRID REASONING MODE

Based on our preliminary findings in Section 2, we use a hybrid reasoning mode, including *Thinking* and *NoThinking*. Generally, many existing reasoning models, such as DeepSeek-R1, adopt the response structure that consists of a long thinking process marked by `<thinking>` and `</thinking>`, followed by the final solution. Our hybrid reasoning mode is defined as follows.

**Thinking** This refers to the default structure for querying language models, in which the model is prompted to generate the detailed thinking process within given special markers. Based on this, the model then derives a solution and generates a final answer presented as its final prediction. This approach encourages the model to reflect, backtrack, and self-verify during the reasoning process.

**NoThinking** This is a concise response structure that includes the final solution and answer without the detailed thinking process within special markers. Instead of using the prompt "`Okay, I think I have finished thinking`" to induce the model to state that it has completed its reasoning (Ma et al., 2025), we explicitly prompt the model to solve the problem without thinking: `<thinking> Okay, I think I can solve it directly. </thinking>`.

We hypothesise that the two modes can be mutually beneficial: while the *Thinking* mode improves performance on complex problems through explicit thinking, the *NoThinking* mode promotes conciseness by reducing verbosity. Specifically, a model trained solely in the *NoThinking* mode may compensate for missing explicit thinking by expanding its reasoning steps, reducing the risk of producing incorrect answers. In other words, *NoThinking* does not simply mean "short" (see Sec. 4.3.) In contrast, when trained jointly with the *Thinking* mode, the model can leverage a compression effect, where improved reasoning ability is distilled into shorter responses. This integration has the potential to yield concise responses without sacrificing accuracy, or may even improve it.

### 3.3 TRAINING WITH GRPO

To train our model efficiently, we adopt the Group Relative Policy Optimisation (GRPO, Shao et al., 2024) along with several improvements recently proposed by the research community. For each problem $q$, the algorithm samples a group of responses $\{o_1, o_2, ..., o_G\}$ from the old policy $\pi_{old}$, and updates the policy $\pi_\theta$ by maximising the following objective:

$$\mathcal{J}_{\text{GRPO}}(\theta) = \mathbb{E}_{q \sim \mathcal{D}, \{o_i\}_{i=1}^G \sim \pi_{\theta_{\text{old}}}(\cdot | q)} \tag{1}$$

$$\frac{1}{G} \sum_{i=1}^{G} \min\left( \frac{\pi_\theta(o_i \mid q)}{\pi_{\theta_{\text{old}}}(o_i \mid q)} A_i, \ \text{clip}\left( \frac{\pi_\theta(o_i \mid q)}{\pi_{\theta_{\text{old}}}(o_i \mid q)}, 1 - \epsilon_{\text{low}}, 1 + \epsilon_{\text{high}} \right) A_i \right)$$

where $\epsilon_{\text{low}}$ and $\epsilon_{\text{high}}$ are hyperparameters for the clipping range of the importance sampling ratio. $A_i$ is the advantage, caculated by the rewards $\{r_1, r_2, ..., r_G\}$ of responses in the same group:

$$A_i = \frac{r_i - \text{mean}(\{r_i\}_{i=1}^G)}{\text{std}(\{r_i\}_{i=1}^G)} \tag{2}$$

Following prior work (Guo et al., 2025), we employ the rule-based reward method, assigning a binary score of 1 for correct final answers and 0 for incorrect ones. We incorporate several enhancements from recent research into the original GRPO algorithm, including (1) removing the KL loss

to effectively unlock the full potential of the policy model without affecting training stability (Hu et al., 2025b); and (2) increasing the upper clip bounds $\epsilon_{high}$ in Eq. 1 while fixing the lower clip bounds $\epsilon_{low}$ to mitigate the entropy convergence problem and encourage the policy to explore diverse solutions (Yu et al., 2025).

## 4 EXPERIMENTS

### 4.1 EXPERIMENTAL SETUP

**Model and Dataset** We select DeepSeek-R1-Distill-Qwen1.5B (R1-Qwen, Guo et al., 2025) as our backbone, given its proven strong performance on mathematical reasoning tasks. We use DeepScaleR-Dataset (Luo et al., 2025b) as our training data, which contains around 40k math problems from various math competitions, including AIME 1983-2023, AMC, Omni-Math (Gao et al., 2025), and STILL (Min et al., 2024). The evaluation is performed on four challenging math benchmarks: MATH500 (Lightman et al., 2024), AMC, AIME 2024, and AIME 2025, which are widely adopted for assessing the models' reasoning abilities (Guo et al., 2025; OpenAI, 2025).[2]

**Baselines** We compare TACLer with several 1.5B baselines trained for long CoT reasoning: (1) **STILL-3** (Chen et al., 2025c), (2) **DeepScaleR** (Luo et al., 2025b), and (3) **FastCuRL** (Song et al., 2025). We also compare TACLer with state-of-the-art 1.5B baselines on efficient math reasoning: (1) **OverThink** (Chen et al., 2025b), (2) **DAST** (Shen et al., 2025), (3) **O1-Pruner** (Luo et al., 2025a), (4) **TLMRE**, (5) ModelMerging (Wu et al., 2025), (6) **AdaptThink** (Zhang et al., 2025), and (7) **AutoThink** (Tu et al., 2025). All models use the DeepScaleR dataset, where methods (1)-(5) are reproduced by Zhang et al. (2025), thereby providing a fair comparison.[3]

### 4.2 MAIN RESULTS

**Reasoning Accuracy and Efficiency in Thinking Mode** Table 1 compares TACLer with several long thinking models using *Thinking* mode. Across four datasets, TACLer achieves the highest accuracy on three of them, and performs slightly lower than DeepScaleR but higher than other models on AIME 2025. Overall, it achieves the highest average accuracy, with an average +11.2% improvement over the base model R1-Qwen. In addition, TACLer reduces reasoning length by 42.7%, substantially outperforming all baselines (e.g., -34.1% for DeepScaleR and -20.2% for FastCuRL). These results highlight that TACLer provides a more effective balance between reasoning accuracy and response length than existing methods in the *Thinking* mode.

| | MATH500 | | AIME 2024 | | AMC | | AIME 2025 | | Average | |
|---|---|---|---|---|---|---|---|---|---|---|
| | ACC | Length ↓ | ACC | Length ↓ | ACC | Length ↓ | ACC | Length ↓ | ΔACC | ΔLength ↓ |
| R1-Qwen$_{Thinking}$ | 81.2 | 4856 | 27.7 | 12306 | 60.8 | 8754 | 21.5 | 12182 | - | - |
| STILL-3 | 83.6 | 3797 | 30.4 | 10605 | 66.3 | 7091 | 24.4 | 10415 | +3.4 | -17.3% |
| DeepScaleR | 87.8 | 3030 | 40.4 | 8565 | 73.8 | 5616 | **31.3** | 8239 | +10.5 | -34.1% |
| FastCuRL | 87.8 | 3894 | 39.8 | 10091 | 73.9 | 6756 | 27.9 | 9723 | +9.6 | -20.2% |
| TACLer$_{Thinking}$ | **88.4** | **3010** | **42.1** | **6868** | **74.6** | **4871** | 30.8 | **6807** | **+11.2** | **-42.7%** |

Table 1: Comparison of model performance (accuracy and response length) in the long thinking mode across four mathematical reasoning datasets. Note that bold numbers indicate the best result for each dataset among different models.

**Reasoning Accuracy and Efficiency in Efficient Mode** Table 2 presents the comparison of TACLer with recent efficient reasoning approaches. The first observation is that TACLer consistently achieves the best overall performance, with the only exception being response length on the MATH500 dataset. Specifically, building upon R1-Qwen model, TACLer boosts the accuracy by over 9%, while simultaneously reducing the average token usage by around half (49.3% and 51.9%). In comparison, efficient reasoning baselines such as AutoThink improve efficiency by reducing token usage, but offer smaller accuracy gains; ModelMerging even suffers from substantial

---

[2]See the Appendix A.1 for more details on training and evaluation.

[3]See the Appendix A.2 for more details on baselines.

|  | MATH500 | | AIME 2024 | | Average | |
|---|---|---|---|---|---|---|
|  | ACC | Length ↓ | ACC | Length ↓ | ΔACC | ΔLength ↓ |
| R1-Qwen$_{Thinking}$ | 81.2 | 4856 | 27.7 | 12306 | - | - |
| R1-Qwen$_{NoThinking}$ | 67.8 | 1069 | 14.8 | 4689 | -13.2 | -69.9% |
| OverThink | 81.2 | 4131 | 28.3 | 11269 | +0.3 | -11.7% |
| DAST | 83.0 | 2428 | 26.9 | 7745 | +0.5 | -43.5% |
| O1-Pruner | 82.2 | 3212 | 28.9 | 10361 | +1.1 | -24.8% |
| TLMRE | 85.0 | 3007 | 29.2 | 8982 | +2.6 | -32.5% |
| ModelMerging | 63.0 | 2723 | 18.1 | 10337 | -13.9 | -30.0% |
| AdaptThink | 86.0 | 2511 | 34.8 | 9279 | +5.9 | -36.4% |
| AutoThink | 83.8 | **2128** | 31.7 | 8167 | +3.3 | -44.9% |
| TACLer$_{NoThinking}$ | **88.2** | 2532 | **39.6** | **6056** | **+9.5** | **-49.3%** |

|  | AMC | | AIME 2025 | | Average | |
|---|---|---|---|---|---|---|
|  | ACC | Length ↓ | ACC | Length ↓ | ΔACC | ΔLength ↓ |
| R1-Qwen$_{Thinking}$ | 60.8 | 8754 | 21.5 | 12182 | - | - |
| R1-Qwen$_{NoThinking}$ | 48.6 | 2264 | 13.3 | 4062 | -10.2 | -70.4% |
| AdaptThink | 67.4 | 5489 | 25.6 | 9117 | +5.4 | -31.2% |
| AutoThink | 66.7 | 4596 | 23.8 | 7647 | +4.1 | -42.4% |
| TACLer$_{NoThinking}$ | **72.7** | **4312** | **27.9** | **5710** | **+9.2** | **-51.9%** |

Table 2: Comparison of model performance (accuracy and response length) in the efficient thinking mode across four mathematical reasoning benchmarks. For MATH500 and AIME 2024, results for all baselines except AutoThink are taken from Zhang et al. (2025). Bold numbers indicate the best result for each dataset among the different models.

performance degradation. On average, across the four datasets, TACLer achieves higher reasoning accuracy and generates shorter responses.

## 4.3 ANALYSIS AND DISCUSSION

**Hybrid Mode Yields Concise Reasoning** We compare TACLer with the pure *NoThinking* model trained in the first stage, as shown in Table 3. TACLer with the hybrid reasoning mode achieves the highest accuracy in the *Thinking* mode, while generating the shortest responses in the *NoThinking* mode. In contrast, Pure-NoThinking achieves higher accuracy than TACLer in the *NoThinking* mode but produces substantially longer responses, failing to support concise reasoning. These results show that the hybrid reasoning mode allows the model to benefit from both strategies, making reasoning under *NoThinking* more concise with no heavy loss in performance.

|  | MATH500 | | AIME 2024 | | AMC | | AIME 2025 | | Average | |
|---|---|---|---|---|---|---|---|---|---|---|
|  | ACC | Length ↓ | ACC | Length ↓ | ACC | Length ↓ | ACC | Length ↓ | ΔACC | ΔLength ↓ |
| R1-Qwen$_{Thinking}$ | 81.2 | 4856 | 27.7 | 12306 | 60.8 | 8754 | 21.5 | 12182 | - | - |
| Pure-NoThinking | 84.0 | 2325 | **32.1** | 8380 | 65.1 | 5223 | 24.8 | 8220 | +3.7 | -39.2% |
| TACLer$_{NoThinking}$ | 82.0 | **1472** | 25.6 | **5882** | 62.1 | **3479** | 22.5 | **5455** | +0.3 | **-59.3%** |
| TACLer$_{Thinking}$ | **84.8** | 3287 | 30.0 | 9168 | **68.2** | 5875 | 25.2 | 8873 | **+4.3** | -29.5% |

Table 3: Comparison of models trained with hybrid reasoning (*NoThinking + Thinking*) versus pure *NoThinking* training after 280 steps of first-stage training. Bold numbers indicate the best result for each dataset among different models.

**Curriculum Learning for More Efficient and Accurate Reasoning** We compare TACLer with Direct-Train, a model which is trained directly using all of the data at once rather than staggered based on difficulty, with all other settings staying the same (Table 4). While Direct-Train tends to generate shorter responses than TACLer under the *NoThinking* mode, the trend is reversed in *Thinking* mode, with TACLer producing more concise outputs compared to Direct-Train. Moreover, TACLer consistently achieves a better performance across both modes, showing an effective balance between conciseness and accuracy.

**TACLer's Performance at Different Stages** Figure 4 shows reasoning accuracy and response length of TACLer at different training stages on two difficulty levels of MATH500 (the original

|  | MATH500 | | AIME 2024 | | AMC | | AIME 2025 | | Average | |
|---|---|---|---|---|---|---|---|---|---|---|
|  | ACC | Length ↓ | ACC | Length ↓ | ACC | Length ↓ | ACC | Length ↓ | ΔACC | ΔLength ↓ |
| R1-Qwen$_{Thinking}$ | 81.2 | 4856 | 27.7 | 12306 | 60.8 | 8754 | 21.5 | 12182 | - | - |
| *NoThinking* Mode | | | | | | | | | | |
| Direct-Train | 87.8 | **2217** | 38.1 | **5974** | 72.1 | **4093** | 26.9 | **5550** | +8.4 | **-53.4%** |
| TACLer | **88.2** | 2532 | **39.6** | 6056 | **72.7** | 4312 | **27.9** | 5710 | **+9.3** | -50.6% |
| *Thinking* Mode | | | | | | | | | | |
| Direct-Train | 88.0 | **2962** | 37.5 | 7085 | 73.0 | 4920 | 27.5 | **6736** | +8.7 | -42.5% |
| TACLer | **88.4** | 3010 | **42.1** | 6868 | **74.6** | **4871** | **30.8** | 6807 | **+11.2** | **-42.7%** |

Table 4: Comparison of curriculum learning and direct training (training from the beginning with all data at once). Bold numbers indicate the best result for each dataset among different models.

5 levels are merged into levels 1–3 and levels 4–5). For reasoning accuracy, we observe a consistent upward trend across the three training stages (see Sec. 3.1.) The largest improvement occurs in the first stage, with higher gains in the *NoThinking* mode. Subsequently, the performance on easier problems (levels 1-3) plateaus, whereas the accuracy on harder ones (levels 4-5) continues to improve. Notably, in the third stage TACLer in *Thinking* mode obtains higher scores on easier problems. We hypothesise that some questions categorised as "easy" by humans may actually be more challenging for the model, and can be successfully solved after the model's capabilities are substantially enhanced through curriculum learning (see examples in Appendix A.3.) When looking at response length, TACLer shows a different trend for *Nothinking* and *Thinking* modes: in the former the length of responses gradually increases with training, while in the latter it gradually decreases.

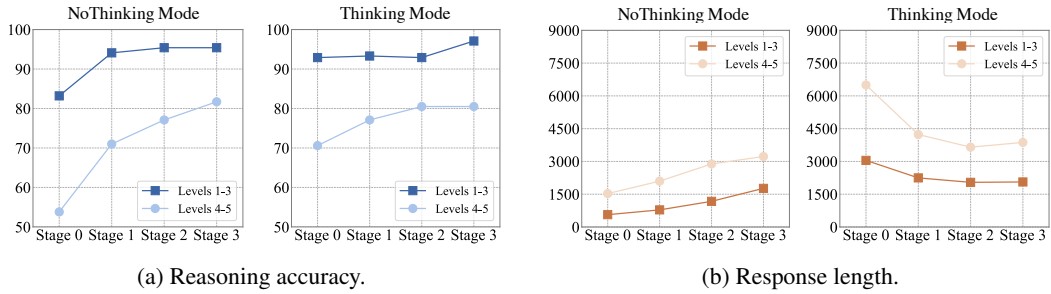

(a) Reasoning accuracy.          (b) Response length.

Figure 4: Comparison of different training stages on different difficulty levels of MATH500 dataset. Note that state 0 represents the base model.

**Computation Cost and Learning Efficiency**
Table 5 summarises the training configurations for different long thinking models to compare their computational requirements. Specifically, DeepScaleR uses progressively increasing context lengths from 8K to 24K and rollout numbers from 8 to 24 in three training stages, while FastCuRL employs a similar strategy with an additional stage. In our settings, TACLer maintains a consistent context length of 8K and rollout of 8 across all three stages. Regarding computational cost, Luo et al. (2025b) indicate that doubling the context window size increases training computation by at least 2 times. At the same time, in our experiments we find that doubling the number of rollouts will roughly increase the computation by about 1 time, indicating that TACLer reduces the required training

| Stage | Context | Batch Size | Step | Rollout |
|---|---|---|---|---|
| DeepScaleR-1.5B-Preview | | | | |
| Stage 1 | 8K | 128 | 1040 | 8 |
| Stage 2 | 16K | 128 | 500 | 16 |
| Stage 3 | 24K | 128 | 210 | 16 |
| FastCuRL-1.5B-Preview | | | | |
| Stage 1 | 8K | 128 | 160 | 8 |
| Stage 2 | 16K | 64 | 590 | 8 |
| Stage 3 | 24K | 64 | 230 | 8 |
| Stage 4 | 16K | 64 | 580 | 16 |
| TACLer | | | | |
| Stage 1 | 8K | 128 | 280 | 8 |
| Stage 2 | 8K | 128 | 350 | 8 |
| Stage 3 | 8K | 128 | 1250 | 8 |

Table 5: Training configuration comparison.

compute by over 50% compared to the baseline models. To validate learning efficiency, in

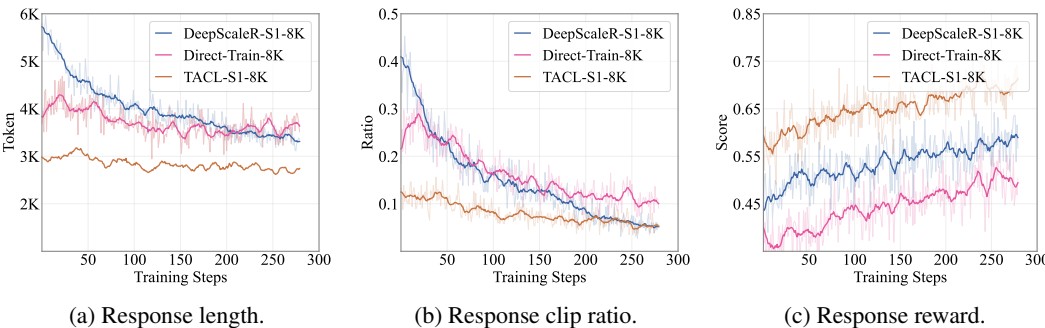

(a) Response length.

(b) Response clip ratio.

(c) Response reward.

Figure 5: Comparison of `TACLer`, Direct-Train, and DeepScaleR-1.5B-Preview in terms of response clip ratio, length, and reward at the first training stage.

Figure 5 we present the response length, clip ratio, and reward for `TACLer` and Direct-Train in the first training stage (280 steps), and compare them with DeepScaleR-1.5B-Preview. Compared to the other two models, `TACLer` generates the shortest responses, indicating faster and more efficient reasoning. This results in a lower response clipping rate, leading to more efficient training. Furthermore, `TACLer` has a higher reward during training. These observations highlight `TACLer`'s efficient learning abilities.

**Dataset Complexity for Curriculum Learning** The model resulting after each training stage is used to curate the difficulty-based dataset to be used in the next stage(s) of tailored curriculum learning. Table 6 summarises the data composition across the three iterative training stages, each consisting of three sets of data of varying difficulty, based on the results of the model R1-Qwen. As training progresses, the proportion of challenging data increases from 0% in the first stage to 34.6% in the second stage and 55.1% in the third stage, while the proportion of easier data decreases accordingly.

|         | G1 (%) | G2 (%) | G3 (%) | Total (#) |
|---------|--------|--------|--------|-----------|
| Stage 1 | 78.1   | 21.9   | 0      | 18110     |
| Stage 2 | 51.1   | 14.3   | 34.6   | 27692     |
| Stage 3 | 35.1   | 9.8    | 55.1   | 40315     |

Table 6: Data complexity statistics for three training stages, as assessed by R1-Qwen (G1: instances with correct answers; G2: instances with complete responses but incorrect answers; G3: instances with incomplete responses, see Section 3.1.)

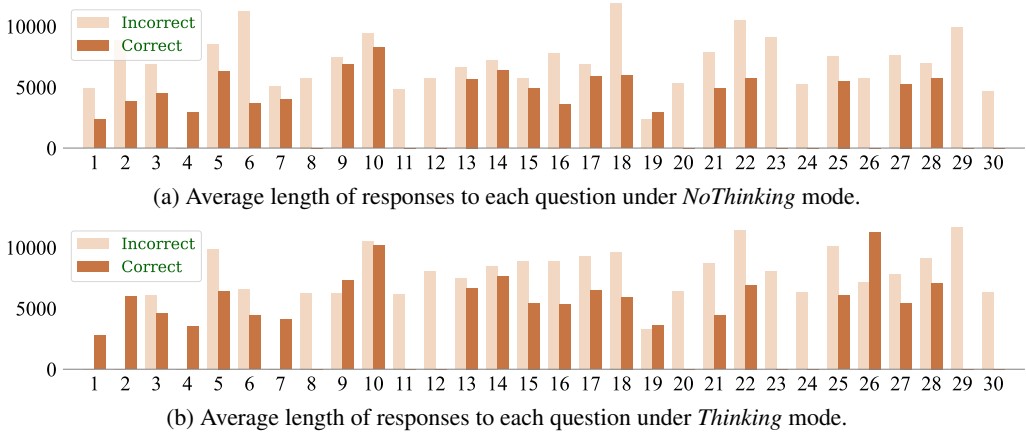

(a) Average length of responses to each question under *NoThinking* mode.

(b) Average length of responses to each question under *Thinking* mode.

Figure 6: Comparison of average response length between correct and incorrect answers. For each problem, we generate 16 responses and compute the average length separately for correct and incorrect responses; when all responses are of one type (correct or incorrect), the average length for the opposite type is set to 0.

**Contrasting Response Lengths in Correct and Incorrect Reasoning** Figure 6 presents the average response length for correct and incorrect answers at the problem level in the *NoThinking* and

*Thinking* modes. Regarding answer correctness, we find that in both reasoning modes, correct responses are shorter than incorrect ones, as the latter often contain more verbose or repetitive content, an observation consistent with previous work (Song et al., 2025). When comparing the responses between the two reasoning modes, correct answers in the *Nothinking* mode tend to have shorter responses than those in the *Thinking* mode, whereas incorrect answers show no clear length trend between the two modes.

## 5 RELATED WORK

**Curriculum Learning** There are two main paradigms of curriculum learning: data-level and model-level (Soviany et al., 2022). Data-level curriculum learning is a method that gradually increases the complexity of the data samples used during the training process (Bengio et al., 2009). This aims to formalise the easy-to-hard learning strategies, thereby mimicking the way humans learn. On the other hand, the model-level method consists in gradually increasing the modelling capacity of the neural model by adding or activating more neural units as the training progresses (Karras et al., 2018; Morerio et al., 2017). In this work, we propose a data-level approach that explicitly incorporates the model's evolving proficiency into the training process. By doing so, we design a tailored curriculum learning framework that can gradually adapt to training data of varying difficulty, thereby facilitating more efficient and effective training of reasoning models.

**RL Training for LLMs** Reinforcement learning has been widely adopted to train LLMs to enhance their abilities to address complex reasoning tasks. Recent works such as OpenAI o1 (OpenAI, 2025) and DeepSeek R1 (Guo et al., 2025) show that large-scale RL training can effectively elicit chain-of-thought (CoT) reasoning, resulting in substantial gains on challenging mathematical reasoning and coding benchmarks. Building on these successes, the community has proposed a series of increasingly sophisticated RL algorithms based on DPO (Rafailov et al., 2023), such as GRPO (Shao et al., 2024), DAPO (Yu et al., 2025), CPPO (Lin et al., 2025), and REINFORCE++ (Hu et al., 2025a). We use the GRPO algorithm together with several recent enhancements.

**Math Reasoning in LLMs** Language models have shown impressive abilities in addressing complex reasoning tasks, thanks to CoT-based strategies which include explicit step-by-step reasoning (Wei et al., 2022). Building upon this foundation, a growing body of research has sought to enhance reasoning performance from various perspectives, such as reasoning consistency (Wang et al., 2023; Zhou et al., 2024; Lai et al., 2025) and multilingual reasoning (Shi et al., 2023; Lai & Nissim, 2024; Chai et al., 2025). Recently, large reasoning models (LRMs), such as OpenAI o1 and DeepSeek R1, have sparked a surge of research on long CoT reasoning, which has significantly improved the mathematical reasoning capabilities of models. Long CoT reasoning involves a detailed iterative exploration and reflection process that is performed within a given problem space by test-time scaling (Chen et al., 2025a). However, recent works have shown that long CoT often induces "overthinking", leading to overly verbose reasoning and thus substantial computational overhead (Sui et al., 2025). To address this problem, a line of work has emerged on efficient reasoning strategies, such as adaptive thinking (Zhang et al., 2025; Tu et al., 2025), length-based rewards (Arora & Zanette, 2025; Shen et al., 2025), and disabling the thinking process (Ma et al., 2025). In this work, we propose an efficient reasoning framework based on tailored curriculum reinforcement learning, showing that the coexistence of *Thinking* and *NoThinking* modes can be mutually beneficial.

## 6 CONCLUSION

We have introduced `TACLer`, a novel tailored curriculum reinforcement learning framework for efficient reasoning. Leveraging staggered training based on progressive data complexity, `TACLer` enables the language model to learn more efficiently and also effectively. Moreover, by adopting a hybrid *Thinking/NoThinking* reasoning mode, `TACLer` shows an effective balance of accuracy and efficiency. Our experiments on the math reasoning task with four benchmarks show that `TACLer` reduces training compute by over 50% and inference token usage by over 42% while improving accuracy by over 9%. A future direction is to further extend our framework to other reasoning tasks or even to general-purpose scenarios. A clear limitation of our method, however, is the additional compute required during inference at each iteration, though the overhead remains relatively small.

# 7 REPRODUCIBILITY STATEMENT

Information about the base model and dataset including training and evaluation sets, can be found in Section 4.1. We also provide the experimental setup in Appendix A.1, containing details on model training and evaluation, such as hyperparameters, hardware, and libraries used. The code and model will be released after publication to ensure reproducibility and support future research and applications.

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

# A APPENDIX

## A.1 TRAINING AND EVALUATION SETTING

We train `TACLer` based on VeRL (Sheng et al., 2025) and perform inference using vLLM (Kwon et al., 2023). Across three training stages, we set the context length to 8K, the batch size to 128, and the learning rate to 1e-6. The hyperparameters $\epsilon_{\text{low}}$ and $\epsilon_{\text{high}}$ are set to 0.2 and 0.28, respectively. `TACLer` is trained on 4 NVIDIA H100 GPUs (94 GB memory each) for approximately 96 hours. We report both the average reasoning accuracy and the response length based on the number of tokens in our evaluation. Following previous work (Zhang et al., 2025) we adopt the sampling method with a context size of 16K, a temperature of 0.6 and a top-$p$ value of 0.95 to generate the response for each question. We sample 16 responses for each question in AIME 2024 and AIME 2025, and report the average results due to their limited dataset size.

## A.2 BASELINES

We compare `TACLer` with state-of-the-art 1.5B baselines on efficient math reasoning.

- **OverThink** (Chen et al., 2025b): a SimPO (Meng et al., 2024) based method, taking the longest response as the negative example and the first two correct reasoning as the positive ones.
- **DAST** (Shen et al., 2025): a SimPO-based method, where preference data is ranked by pre-sampled responses using a length-based reward function.
- **O1-Pruner** (Luo et al., 2025a): it builds a baseline via pre-sampling and guides the model to generate more concise reasoning under accuracy constraints.
- **TLMRE** (Arora & Zanette, 2025): a method that incorporates a length-based penalty in on-policy RL training.
- **ModelMerging** (Wu et al., 2025): it takes a weighted average of the weights of the reasoning model and the non-reaosning model.
- **AdaptThink** (Zhang et al., 2025): a method trained with a constrained objective to encourage the model to reason in the NoThinking way.
- **AutoThink** (Tu et al., 2025): an RL-based approach optimizes models to invoke long inferences only when necessary.

All above methods use DeepScaleR dataset, where methods (1)-(5) are reproduced by Zhang et al. (2025), thereby having a fair comparison. We also compare our method with several 1.5B baselines trained for long CoT reasoning in the *Thinking* mode.

- **STILL-3** (Chen et al., 2025c): a method to enhance the model's slow thinking through RL training.
- **DeepScaleR** (Luo et al., 2025b): an iterative training method that iteratively increases the context length from 8K to 24K, allowing the model to improve the quality of long CoT.
- **FastCuRL** (Song et al., 2025): a curriculum RL framework with progressively extended context, in which the training data complexity is categorised based on input length.

## A.3 CASE STUDY

Figure 7 presents a level-2 problem from the MATH500 dataset, which is considered relatively easy by humans but proves more challenging for language models. The response generated by R1-Qwen under the *Thinking* mode produces an incorrect answer, while `TACLer` answers correctly. Figures 8 and 9 present a math problem at difficulty level 3, comparing outputs of the base model R1-Qwen and our model `TACLer` under two reasoning modes: *Thinking* and *NoThinking*. For R1-Qwen, the output under the *NoThinking* mode is short but the final answer is incorrect, while the final answer in the *Thinking* mode is correct but the output is too long. In contrast, `TACLer` produces shorter outputs with correct final answers in both modes, with the output under the *NoThinking* mode being about half the length of that under the *Thinking* mode. These cases highlight the differences in reasoning performance between the two models.

**Question (MATH500):** Evaluate\n\\[\\sin (\\arcsin 0.4 + \\arcsin 0.5) \\cdot \\sin (\\arcsin 0.5 - \\arcsin 0.4).\\]

**Difficulty Level**: 2

**Answer**: \\frac{9}{100}

**DeepSeek-R1-Distill-Qwen-1.5B (*Thinking* Mode):** 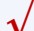

<think>\n Okay, so I have this problem here: I need to evaluate the expression \\(\\sin (\\arcsin 0.4 + \\arcsin 0.5) \\cdot \\sin (\\arcsin 0.5 - \\arcsin 0.4)\\). Hmm, that looks a bit complicated, but maybe I can break it down step by step.\n\nFirst, I remember that there are some trigonometric identities that might help simplify this. I recall the sine addition and subtraction formulas. Let me write them down:\n\n1. \\(\\sin(A + B) = \\sin A \\cos B + \\cos A \\sin B\\)\n2. \\(\\sin(A - B) = \\sin A \\cos B - \\cos A \\sin B\\)\n\n So, if I let \\(A = \\arcsin 0.4\\) and \\(B = \\arcsin 0.5\\), then the expression becomes \\(\\sin(A + B) \\cdot \\sin(A - B)\\). That seems manageable.\n\nLet me compute each part separately. Let me denote \\(A = \\arcsin 0.4\\), which means that \\(\\sin A = 0.4\\). Similarly, \\(B = \\arcsin 0.5\\), so \\(\\sin B = 0.5\\). Since both arcsin functions are in the range \\([- \\pi/2, \\pi/2]\\), both A and B are in the first quadrant, so their cosines will be positive.\n **[...] (omit 7105 tokens)** \n\n**Final Answer**\n\n\boxed{-\\dfrac{9}{100}}\n</think>\n\nTo evaluate the expression \\(\\sin (\\arcsin 0.4 + \\arcsin 0.5) \\cdot \\sin (\\arcsin 0.5 - \\arcsin 0.4)\\), we start by letting \\(A = \\arcsin 0.4\\) and \\(B = \\arcsin 0.5\\). This transforms the expression into \\(\\sin(A + B) \\cdot \\sin(A - B)\\).\n\nUsing the sine addition and subtraction formulas:\n\\[\n\\sin(A + B) = \\sin A \\cos B + \\cos A \\sin B\n\\]\n\\[\n\\sin(A - B) = \\sin A \\cos B - \\cos A \\sin B\n\\]\n\nWe know:\n- \\(\\sin A = 0.4\\)\n- \\(\\sin B = 0.5\\)\n- \\(\\cos A = \\sqrt{1 - (0.4)^2} = \\sqrt{0.84}\\)\n- \\(\\cos B = \\sqrt{1 - (0.5)^2} = \\sqrt{0.75}\\)\n\nUsing the identity for the product of sines:\n\\[\n\\sin(A + B) \\cdot \\sin(A - B) = \\sin^2 A \\cos^2 B - \\cos^2 A \\sin^2 B\n\\]\n\nSubstituting the known values:\n\\[\n\\sin^2 A = (0.4)^2 = 0.16\n\\]\n\\[\n\\cos^2 B = (\\sqrt{0.75})^2 = 0.75\n\\]\n\\[\n\\cos^2 A = (\\sqrt{0.84})^2 = 0.84\n\\]\n\\[\n\\sin^2 B = (0.5)^2 = 0.25\n\\]\n\nCalculating each term:\n\\[\n\\sin^2 A \\cos^2 B = 0.16 \\times 0.75 = 0.12\n\\]\n\\[\n\\cos^2 A \\sin^2 B = 0.84 \\times 0.25 = 0.21\n\\]\n\nSubtracting these results:\n\\[\n0.12 - 0.21 = -0.09\n\\]\n\nThus, the value of the expression is:\n\\[\n\\boxed{-\\dfrac{9}{100}}\n\\] **(8508 tokens in total)**

**TACLer-1.5B (*Thinking* Mode):** √

<think>\n Okay, so I have this trigonometric expression to evaluate: \\[\\sin (\\arcsin 0.4 + \\arcsin 0.5) \\cdot \\sin (\\arcsin 0.5 - \\arcsin 0.4).\\] Hmm, that looks a bit complicated at first glance, but maybe I can break it down step by step.\n\nFirst, I remember that there are some trigonometric identities that relate sine and arcsine functions. Specifically, the sine of a sum and difference of two angles. Maybe I can use those identities here.\n\nLet me recall the sine addition and subtraction formulas:\n\n1. \\(\\sin(A + B) = \\sin A \\cos B + \\cos A \\sin B\\)\n2. \\(\\sin(A - B) = \\sin A \\cos B - \\cos A \\sin B\\)\n\nSo, the expression is the product of two sine terms:\n\n\\[\\sin(\\arcsin 0.4 + \\arcsin 0.5)] \\cdot [\\sin(\\arcsin 0.5 - \\arcsin 0.4)]\n\\]\n\n Maybe I can compute each sine term separately and then multiply them together.\n\nLet me denote:\n\nLet \\( \\alpha = \\arcsin 0.4 \\) and \\( \\beta = \\arcsin 0.5 \\).\n\nSo, the expression becomes:\n\n\\[\n\\sin(\\alpha + \\beta) \\cdot \\sin(\\beta - \\alpha)\n\\]\n\nHmm, interesting. Now, I can use the sine addition and subtraction formulas on each sine term.\n\nFirst, let's compute \\( \\sin(\\alpha + \\beta) \\):\n\nUsing the addition formula:\n\n\\[\n\\sin(\\alpha + \\beta) = \\sin \\alpha \\cos \\beta + \\cos \\alpha \\sin \\beta\n\\]\n\nSimilarly, \\( \\sin(\\beta - \\alpha) \\) can be expanded using the subtraction formula:\n\n\\[\n\\sin(\\beta - \\alpha) = \\sin \\beta \\cos \\alpha - \\cos \\beta \\sin \\alpha\n\\]\n\nSo, the original expression becomes:\n\n\\[[\\sin \\alpha \\cos \\beta + \\cos \\alpha \\sin \\beta] \\cdot [\\sin \\beta \\cos \\alpha - \\cos \\beta \\sin \\alpha]\n\\]\n\nLet me write that down:\n\n\\[(\\sin \\alpha \\cos \\beta + \\cos \\alpha \\sin \\beta)(\\sin \\beta \\cos \\alpha - \\cos \\beta \\sin \\alpha)\n\\]\n\n **[...] (omit 2353 tokens)** I didn't need to compute all those square roots and stuff. Just applying the identity.\n\nTherefore, the answer is 9/100.\n\n **Final Answer**\n\\boxed{\\dfrac{9}{100}}\n</think>\n\nTo evaluate the expression \\(\\sin (\\arcsin 0.4 + \\arcsin 0.5) \\cdot \\sin (\\arcsin 0.5 - \\arcsin 0.4)\\), we can use trigonometric identities and properties of inverse sine functions.\n\nFirst, let \\(\\alpha = \\arcsin 0.4\\) and \\(\\beta = \\arcsin 0.5\\). The expression becomes:\n\\[\n\\sin(\\alpha + \\beta) \\cdot \\sin(\\beta - \\alpha)\n\\]\n\nUsing the identity for the product of sines, we recognize that:\n\\[\n\\sin(\\alpha + \\beta) \\cdot \\sin(\\beta - \\alpha) = \\sin^2(\\beta) - \\sin^2(\\alpha)\n\\]\n\nSince \\(\\alpha = \\arcsin 0.4\\) and \\(\\beta = \\arcsin 0.5\\), we have:\n\\[\n\\sin(\\alpha) = 0.4 \\quad \\text{and} \\quad \\sin(\\beta) = 0.5\n\\]\n\nThus, we can compute:\n\\[\n\\sin^2(\\beta) - \\sin^2(\\alpha) = (0.5)^2 - (0.4)^2 = 0.25 - 0.16 = 0.09 = \\frac{9}{100}\n\\]\n\nTherefore, the final answer is:\n\\[\n\\boxed{\\dfrac{9}{100}}\n\\] **(3395 tokens in total**

Figure 7: Example of a MATH500 problem at difficulty level 2 and the responses generated by R1-Qwen and `TACLer` under the *Thinking* mode.

**Question (MATH500):** How many positive two-digit integers are factors of both 100 and 150?

**Difficulty Level:** 3

**Answer:** 3

**DeepSeek-R1-Distill-Qwen-1.5B (*NoThinking* Mode):** ✗

**\<think\>**\nOkay, I think I can solve it directly.**\</think\>**\n\n**Problem:** \nHow many positive two-digit integers are factors of both 100 and 150?\n\n**Solution:**\n\nTo find the positive two-digit integers that are factors of both 100 and 150, we'll follow these steps:\n\n1. **Find the Greatest Common Factor (GCF) of 100 and 150.** \n  - **Prime Factorization of 100:** \n     \\(100 = 2^2 \\times 5^2\\)\n  - **Prime Factorization of 150:** \n     \\(150 = 2 \\times 3 \\times 5^2\\)\n  - **Identify Common Factors:** \n  The common prime factors are \\(2\\) and \\(5^2\\). \n       So, \\(GCF = 2 \\times 5^2 = 50\\).\n\n2. **Determine the Two-Digit Factors of 50.** \n  - **List the Factors of 50:** \n    1, 2, 5, 10, 25, 50 \n  - **Filter for Two-Digit Integers:** \n    From the list, the two-digit factors are 10 and 25.\n\n3. **Count the Two-Digit Factors:** \n  There are **2** positive two-digit integers (10 and 25) that are factors of both 100 and 150.\n\n**Final Answer:** \n\\boxed{2}. **(372 tokens)**

**DeepSeek-R1-Distill-Qwen-1.5B (*Thinking* Mode):** √

**\<think\>**\nAlright, so I need to figure out how many positive two-digit integers are factors of both 100 and 150. Hmm, okay. Let me break this down step by step. \n\nFirst, I remember that if a number is a factor of both 100 and 150, it must be a common factor of these two numbers. So, I need to find all the common factors of 100 and 150 that are two-digit numbers. That makes sense.\n\nMaybe I should start by finding the factors of each number individually and then see where they overlap. That sounds like a plan. Let me start with 100. \n\nThe factors of 100 are all the numbers that can divide 100 without leaving a remainder. Let me list them out. Starting from 1:\n\n1, 2, 4, 5, 10, 20, 25, 50, 100. \n\nWait, let me check that. 1 times 100 is 100, so that's a factor. 2 times 50 is 100, so 2 and 50 are factors. Then 4 times 25 is 100, so 4 and 25. 5 times 20 is 100, so 5 and 20. Then 10 times 10 is 100, so 10 and 10. So yeah, those are all the factors of 100. \n\nNow, let me do the same for 150. The factors of 150 are a bit more, I think. Let me see:\n\n1, 2, 3, 5, 6, 10, 15, 25, 30, 50, 75, 150. \n\nLet me verify. 1 times 150 is 150, 2 times 75 is 150, 3 times 50 is 150, 5 times 30 is 150, 6 times 25 is 150, 10 times 15 is 150. Then 15 times 10 is 150, 25 times 6 is 150, 30 times 5 is 150, 50 times 3 is 150, 75 times 2 is 150, and 150 times 1 is 150. So, that seems comprehensive.\n\nNow, I need to find the common factors of both 100 and 150. So, looking at the factors of 100: 1, 2, 4, 5, 10, 20, 25, 50, 100. And factors of 150: 1, 2, 3, 5, 6, 10, 15, 25, 30, 50, 75, 150.\n\nComparing these two lists, the common factors are the numbers that appear in both. So, let's see:\n\n1 is common, 2 is common, 5 is common, 10 is common, 25 is common, 50 is common. Wait, what about 150? It's only in the factors of 150, not in 100. Similarly, 75 is only in 150, not in 100. So, the common factors are 1, 2, 5, 10, 25, 50. \n\nBut hold on, the question is asking for positive two-digit integers. So, two-digit numbers are from 10 to 99 inclusive. Looking at the common factors, we have 10, 25, and 50. \n\nWait, let me double-check that. 1 is a single-digit number, so it doesn't count. 2 is also single-digit. 5 is single-digit, 10 is two-digit, 25 is two-digit, 50 is two-digit. So, the common two-digit factors are 10, 25, and 50.\n\nIs that all? Let me make sure I didn't miss any. Looking back at the factors of 100: 1, 2, 4, 5, 10, 20, 25, 50, 100. And factors of 150: 1, 2, 3, 5, 6, 10, 15, 25, 30, 50, 75, 150. \n\nSo, the common factors are indeed 1, 2, 5, 10, 25, 50. So, within these, the two-digit numbers are 10, 25, and 50. \n\nBut wait a second, 50 is a two-digit number, yes, but 100 is three-digit, so it's excluded. So, 10, 25, 50 are the common two-digit factors.\n\nIs there any chance I missed any other common factors? Let me think. 100 is 2^2 * 5^2, and 150 is 2 * 3 * 5^2. So, the greatest common divisor (GCD) of 100 and 150 would be the product of the lowest powers of the common prime factors. So, GCD is 2^1 * 5^2 = 2 * 25 = 50. So, the GCD is 50. That means the common factors are all the factors of 50.\n\nWait, hold on, factors of 50. Let me list the factors of 50: 1, 2, 5, 10, 25, 50. So, yeah, that's consistent with what I found earlier. So, the common factors are exactly the factors of 50. So, the two-digit ones are 10, 25, 50.\n\nTherefore, the positive two-digit integers that are factors of both 100 and 150 are 10, 25, and 50. So, that's three numbers. **[…] (omit 1346 tokens)** \n\nI think that's solid. I don't see any mistakes in the reasoning. So, I'm confident that the answer is 3.\n\n**Final Answer**\nThe number of positive two-digit integers that are factors of both 100 and 150 is \\boxed{3}.\n**\</think\>**\n\nTo determine how many positive two-digit integers are factors of both 100 and 150, we start by finding the factors of each number.\n\nThe factors of 100 are: 1, 2, 4, 5, 10, 20, 25, 50, 100.\n\nThe factors of 150 are: 1, 2, 3, 5, 6, 10, 15, 25, 30, 50, 75, 150.\n\nNext, we identify the common factors of both sets. These common factors are: 1, 2, 5, 10, 25, 50.\n\nWe are specifically interested in the two-digit integers among these common factors. The two-digit numbers are 10, 25, and 50.\n\nThus, the number of positive two-digit integers that are factors of both 100 and 150 is \\(\\boxed{3}\\). **(3114 tokens in total)**

Figure 8: Example of a MATH500 problem and the responses generated by R1-Qwen under two reasoning modes.

**Question (MATH500):** How many positive two-digit integers are factors of both 100 and 150?
**Difficulty Level**: 3
**Answer:** 3

**TACLer-1.5B (*NoThinking* Mode):** 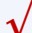
**<think>**\nOkay, I think I can solve it directly.**</think>**\n\n Alright, so I need to find how many positive two-digit integers are factors of both 100 and 150. Hmm, that sounds like I need to find the common factors of 100 and 150 that are two-digit numbers. Let me break this down step by step.\n\nFirst, maybe I should find all the factors of 100 and all the factors of 150, and then see which ones they have in common, and among those, how many are two-digit numbers. Alternatively, maybe it's easier to find the greatest common factor (GCF) of 100 and 150 first and then find all two-digit factors of that GCF.\n\nWait, that might be a good approach. Let me try that.\n\nSo, first, let me find the GCF of 100 and 150.\n\nTo find the GCF, I can use the prime factorization method.\n\nLet's factorize 100:\n100 can be broken down into prime factors:\n100 = 2^2 * 5^2\n\nNow, factorize 150:\n150 = 2 * 3 * 5^2\n\nSo, the GCF is the product of the lowest powers of the common primes. **[...] (omit 660 tokens)** \n\nFactors of 150: 1, 2, 3, 5, 6, 10, 15, 25, 30, 50, 75, 150.\n\nCommon factors: 1, 2, 5, 10, 25, 50.\n\nSo, the common two-digit factors are 10, 25, 50.\n\nHence, there are 3 such numbers.\n\nTherefore, the answer is 3.\n\nWait, just to make sure I didn't miss any, let me list the two-digit numbers that are factors of both 100 and 150.\n\nFrom the list of common factors: 10, 25, 50.\n\nYep, those are the only two-digit numbers.\n\nSo, the number of positive two-digit integers that are factors of both 100 and 150 is 3.\n\n**Final Answer**\nThe number of positive two-digit integers that are factors of both 100 and 150 is \\boxed{3}. **(1203 tokens in total)**

**TACLer-1.5B (*Thinking* Mode):** 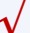
**<think>**\nOkay, so I have this problem: \"How many positive two-digit integers are factors of both 100 and 150?\" Hmm, let me try to figure this out step by step. I remember that when we're looking for common factors of two numbers, we can use the concept of the greatest common divisor (GCD). The GCD of two numbers is the largest number that divides both of them without leaving a remainder. But here, we're not just looking for the GCD, but specifically the two-digit factors of that GCD. Wait, maybe I need to clarify that.\n\nFirst, let me find the factors of 100 and 150 separately and then see what they have in common. But that might take too long because 100 and 150 are not too large, but it's better to use the GCD method.\n\nSo, let's find the GCD of 100 and 150. To do that, I can use the prime factorization method. Let me break down both numbers into their prime factors.\n\nStarting with 100: 100 divided by 2 is 50, divided by 2 again is 25, which is 5 squared. So, 100 is 2^2 * 5^2.\n\nNow for 150: 150 divided by 2 is 75, then 75 divided by 3 is 25, which is 5 squared. So, 150 is 2 * 3 * 5^2.\n\nTo find the GCD, I take the lowest power of all common prime factors. The common primes here are 2 and 5. The lowest power of 2 is 2^1 and the lowest power of 5 is 5^2. So, GCD = 2^1 * 5^2 = 2 * 25 = 50.\n\nSo, the greatest common divisor of 100 and 150 is 50. That means the largest number that divides both 100 and 150 is 50. But the question is asking for two-digit integers that are factors of both. So, we need to find all two-digit numbers that are factors of 50.\n\nWait, hold on. If the GCD is 50, then the factors of both 100 and 150 are exactly the factors of 50. So, now I need to list all the two-digit factors of 50.\n\nWhat are the factors of 50? Let me list them: 1, 2, 5, 10, 25, 50. So, these are the factors of 50. Now, among these, the two-digit numbers are 10, 25, and 50. But wait, 10, 25, and 50. But 50 is a two-digit number? Wait, 50 is a two-digit number, yes.\n\nWait, but hold on. The question is asking for positive two-digit integers that are factors of both 100 and 150. So, if 50 is a factor of both, then 50 is a two-digit number that satisfies the condition. Similarly, 10 and 25 are two-digit numbers.\n\nSo, let me list them: 10, 25, 50. Are these all the two-digit factors of 50? Let's see: 10 is 10, 25 is 25, 50 is 50. Are there any other two-digit numbers that are factors of 50? Let me check.\n\n50 divided by 5 is 10, which is two-digit. 50 divided by 10 is 5, which is single-digit. 50 divided by 25 is 2, which is single-digit. 50 divided by 50 is 1, which is single-digit. So, the only two-digit factors are 10, 25, and 50. So that's three numbers. **[...] (omit 945 tokens)** \n\nYes.\n\nSo, all three numbers are valid.\n\nTherefore, the number of positive two-digit integers that are factors of both 100 and 150 is 3.\n\n**Final Answer**\nThe number of positive two-digit integers that are factors of both 100 and 150 is \\boxed{3}.\n**</think>**\n\nTo determine how many positive two-digit integers are factors of both 100 and 150, we start by finding the greatest common divisor (GCD) of 100 and 150 using their prime factorizations.\n\nThe prime factorization of 100 is:\n\\[ 100 = 2^2 \\times 5^2 \\]\n\nThe prime factorization of 150 is:\n\\[ 150 = 2 \\times 3 \\times 5^2 \\]\n\nThe GCD is found by taking the lowest power of all common prime factors:\n\\[ \\text{GCD} = 2^1 \\times 5^2 = 2 \\times 25 = 50 \\]\n\nNext, we list the factors of 50, which are:\n\\[ 1, 2, 5, 10, 25, 50 \\]\n\nWe are interested in the two-digit factors of 50. These are:\n\\[ 10, 25, 50 \\]\n\nThus, the number of positive two-digit integers that are factors of both 100 and 150 is:\n\\[\\boxed{3}\\]. **(2227 tokens in total)**

Figure 9: Example of a MATH500 problem and the responses generated by our model `TACLer` under two reasoning modes.

