# OpenReview forum: "TACLer: Tailored Curriculum Reinforcement Learning for Efficient Reasoning"
_ICLR.cc/2026/Conference — Submitted to ICLR 2026_

### Official Review · Reviewer_kYe1 · 2025-10-27

**Soundness:** 2
**Presentation:** 3
**Contribution:** 2
**Rating:** 2
**Confidence:** 4

**Summary:**

Reinforcement Learning (RL) models trained with long Chain of Thought (CoT) reasoning often struggle with costly and excessive "thinking" due to redundant tokens. To mitigate this, the authors propose TACLer, a novel approach that categorizes data by difficulty and employs a hybrid strategy, dynamically deciding whether to engage in complex thought processes or solve problems directly. The model was trained using a modified GRPO in a continual learning framework. When applied to mathematical datasets, this method demonstrated improved accuracy while significantly reducing the length of CoT reasoning.

**Strengths:**

- The paper presents interesting, although already well-known, findings. The use of a hybrid thinking model, which employs a long chain of thought for complex cases and direct answers for simpler ones, is particularly interesting. Ablation studies demonstrate the benefits of this strategy compared to either extreme.

- The paper is well-written and effectively highlights the desired accuracy-efficiency Pareto gains.

**Weaknesses:**

- The paper's main idea that a hybrid model is better than "thinking" or "non-thinking" ones doesn't have enough experimental proof. The experiments only used one small model (1.5B, which isn't big enough to say for sure) and one domain (math). We don't know if this approach would work for bigger models or other areas like coding. Moreover, using the DeepScaleR dataset is a bit iffy because it might have some of the same data as the AMC and AIME datasets. It's not clear if the authors checked for duplicates or overlaps.

- A significant concern with the current setup is during the inference process. There's no automatic method to determine whether a query requires "thinking mode" or not. If a user runs it twice, which output should be selected is not clear. It would be valuable to establish an upper bound for direct thinking and non-thinking models (by taking a union of their accuracies) and compare it against TACLer. This would help in assessing the limitations of hybrid models. Furthermore, it appears that TACLer's "no-thinking mode" often incurs higher token costs than comparable modes in other models, such as Qwen. The reason for this discrepancy is not clear.

- It's tough to argue that how hard a problem is directly relates to how much it costs in tokens. For example, a simple multi-step math word problem would use more tokens than just multiplying two three-digit numbers. But TACLer would say the word problem needs "thought" and the multiplication doesn't, which feels a bit off.

- The paper really talks up the cost savings of curriculum learning, but it doesn't get into the training costs (like FLOPS, GPU hours, etc.). Every step of this process needs a full dataset inference and then difficulty labeling. This seems like a very expensive process. Also, Table 6 brings up using R1 Qwen for difficulty assessment. The paper should explain what this model is, why it was picked, and why they didn't just use the generation length method they mentioned earlier instead.

- [MINOR] Why was KL and clipping removed from GRPO? Was the approach tested with KL and clipping and was that not working well?

**Questions:**

1. Checking duplicates in DeepScalerR dataset and making sure there are no train test overlaps.

2. How TACLer works during inference is not clear? If the user needs to decide which mode to choose, it seems like the best bet is to always choose the thinking mode and then this hybrid approach converges to a thinking model.
3. More discussion on the training cost is needed to understand the train costs compared to other baselines.
4. Can the approach be generalized beyond math? And can this be scaled to larger models or to other model families?
5. Table 6 can also be reported in terms of absolute token costs instead of percentage. That would be more clear.
6. Why were tweaks made to the GRPO algorithm? What was the reasoning behind it?
7. How are the token costs calculated? Which tokenizer was used?

---

> ### Author Response · Authors · 2025-11-20
> **[1] Response to Reviewer kYe1**
>
> We thank the reviewer for the careful reading and constructive feedback. Below we address each of the concerns in turn.
>
> **Experimental scope and model scale**
> We acknowledge the limited scale of our experiments. Large-scale RL training beyond a few billion parameters is often infeasible in academic settings. Our goal is to provide controlled insights into selective math reasoning rather than to compete at an industrial scale. We believe even at 1.5B parameters, the observed accuracy–efficiency trade-offs reveal general principles that can guide future scaling efforts.
>
> **Dataset overlap and duplicate checking**
> We appreciate the reviewer’s concern. The original DeepScaleR construction already employs RAG with embeddings and overlap checking to eliminate duplicate problems. We additionally check for overlaps between training and test sets, confirming that no data contamination occurs in experiments.
>
> **There's no automatic method to determine whether a query requires "thinking mode" or not.**
> As stated in lines 52-57, we argue that adaptive reasoning strategies tend to constrain users' options, and therefore a more effective paradigm should enable the coexistence of Thinking and Nothinking modes within a model. In other words, if users require high accuracy or detailed thinking traces, then Thinking mode should be used; if higher efficiency is prioritised, then a "non-thinking mode" is preferrable.
>
> **Upper bound for both modes**
> We thank the reviewer for this insightful suggestion. Following the comment, we introduce an 'oracle union' to estimate the theoretical performance ceiling, as shown in the table below. Across four datasets, TACLer achieves over 86% of this upper bound’s accuracy, demonstrating that the hybrid design effectively balances performance and efficiency with minimal loss to the theoretical maximum.
>
> |Mode                          | MATH500  | AIME2024  |   AMC     | AIME2024  |   Relative to upper bound(%)|
> |------------------------------|----------------|----------------|----------------|-----------------|-----------------|
> | TACLer-Upper bound |     91.4       |       52.1      |     80.8      |      33.8       |  -                  |
> | TACLer-NoThinking   |     88.2       |       39.6      |     72.7       |     27.9        |   86.3           |
> | TACLer-Thinking        |     88.4       |       42.1      |     74.6       |     30.8        |   90.2           |
>
>
> **TACLer's "no-thinking mode" often incurs higher token costs than the base mode R1-Qwen**
> Thanks for pointing this out. We agree that this aspect is not sufficiently discussed in the Results and Analysis section. As noted in the Methodology, the NoThinking mode initially underperforms and tends to compensate for the lack of explicit reasoning by slightly extending its response length, thereby reducing the risk of incorrect answers. Despite this moderate increase, NoThinking still produces shorter outputs than Thinking mode, while achieving noticeably better accuracy than the base model. We will include the corresponding discussion in the revision.
>
> **Relationship between difficulty and token cost**
> We agree that token cost does not perfectly correspond to problem difficulty, as illustrated by the reviewer’s example. However, for most reasoning-focused tasks, especially the complex math word problems studied in this work, solving the problem typically requires multi-step understanding, intermediate reasoning, and constraint handling. These processes naturally lead to longer reasoning traces and therefore produce a strong correlation between difficulty and token usage in practice. We will clarify this point explicitly in the revised version.
>
> **Computational cost**
> We appreciate the reviewer’s concern regarding this point. We do report training compute in the submission (line 709): TACLer is trained on 4 NVIDIA H100 GPUs for approximately 96 hours. Since not all papers provide this information, we follow prior work and compare computation costs using context length, training steps, and rollout numbers, as reported in Table 5. This allows for a consistent and fair evaluation of training efficiency across different methods. Additionally, Deepscaler reports that their full training run requires 3,800 A100 GPU hours. NVIDIA states that the H100 provides a 2–3× speedup over the A100. This indicates that our method substantially reduces computational requirements. We will add the above information to the revision.
>
> **Data labelling cost (inference)**
> Regarding the cost of problem-difficulty labelling, we note in the Conclusion that the overhead is relatively small. Concretely, A full dataset inference requires around 5 hours on a single H100 GPU, and after the first and second training rounds, this time is further reduced due to the significant improvement in TACLer’s reasoning efficiency. We will further stress  these details in the revision.

---

> ### Author Response · Authors · 2025-11-20
> **[2] Response to Reviewer kYe1**
>
> **Why choose R1-Qwen for difficulty assessment**
> Thanks for pointing this out. Our data difficulty labelling is indeed based on generation length and final answer correctness, using the iterative model to construct a tailored curriculum. In Table 6, the “Total (#)” column reflects how data selection changes based on this mechanism. To provide an additional objective assessment of the overall difficulty progression, we chose the base model R1-Qwen to re-evaluate the data, confirming that training examples generally increase in difficulty as intended.
>
> **Why not use the absolute cost of the token instead of a percentage**
> Since we set the maximum length to 8k tokens, models may fail to generate full solutions for some questions. Therefore, we continue to use the data difficulty labelling method to report the proportion of each group.
>
> **Tokeniser**
> We use the tokeniser of DeepSeek-R1-Distill-Qwen1.5B, which is our base model. Therefore, all token counts reported in the paper are computed consistently using this tokeniser.
>
> We thank the reviewer again for the constructive insights and feedback, which surely help to improve the paper’s clarity and contribution.

---

### Official Review · Reviewer_hVt8 · 2025-10-29

**Soundness:** 2
**Presentation:** 3
**Contribution:** 2
**Rating:** 4
**Confidence:** 3

**Summary:**

# Summary

This paper introduces **TACLer** (Tailored Curriculum Reinforcement Learning), a framework designed to improve both training efficiency and reasoning efficiency for large language models on mathematical reasoning tasks. The key contributions are:

1. **Tailored Curriculum Learning**: Unlike conventional curriculum learning that uses fixed difficulty metrics, TACLer dynamically assesses problem difficulty based on the model's current proficiency (measured by pass rate), organizing training data into progressive stages that adapt to what the model specifically needs to learn.

2. **Hybrid Reasoning Mode**: The framework incorporates both Thinking mode (with explicit `<thinking>` tags for long chain-of-thought reasoning) and NoThinking mode (concise responses without explicit thinking markers), allowing flexible balance between accuracy and computational efficiency.

3. **Training Efficiency**: By maintaining a consistent 8K context window across all training stages (vs. 8K→24K in baselines), TACLer reduces training compute by over 50% compared to models like DeepScaleR-1.5B.

4. **Strong Empirical Results**: On four math benchmarks (MATH500, AMC, AIME 2024/2025), TACLer achieves +9-11% accuracy improvement over the base model while reducing average token usage by 42-51%, outperforming both long-thinking baselines and efficient reasoning methods.

The paper demonstrates that curriculum learning tailored to model proficiency, combined with hybrid reasoning modes, can simultaneously improve learning efficiency (faster training), reasoning efficiency (shorter responses), and accuracy on complex mathematical reasoning tasks.

**Strengths:**

# Strengths

The paper makes meaningful contributions in both methodology and practical impact. **Originality**: TACLer introduces a novel model-adaptive curriculum learning approach that dynamically adjusts training difficulty based on the model's actual pass rate rather than arbitrary metrics (e.g., input length), effectively addressing the high truncation rates (>40%) observed in prior work. **Quality and Significance**: The framework achieves substantial practical improvements—reducing training compute by over 50% while improving accuracy by 9-11% and reducing inference tokens by 42-51% across four challenging math benchmarks. The experimental evaluation is comprehensive, comparing against 10+ recent baselines covering both long-thinking and efficient reasoning methods.

**Weaknesses:**

## 1. Writing Quality Issue (Minor)
**Line 48**: Repetitive phrasing - "applying techniques such as such as length-based rewards" contains duplicate "such as". This should be corrected to "applying techniques such as length-based rewards".

## 2. Insufficient Explanation of Efficiency Gains in Thinking Mode
**Section 4.2 & 3.3**: The paper reports significant response length reduction (-42.7%) in Thinking mode (Table 1), but fails to explain the mechanism behind this improvement. The training methodology (Section 3.3) only describes using GRPO with two tricks (removing KL loss and increasing upper clip bounds), neither of which explicitly targets response length control. Since GRPO itself suffers from overthinking issues (as acknowledged in Section 2), the source of efficiency gains remains unclear. The paper needs to:
- Explicitly explain which component(s) of the framework lead to shorter responses
- Provide ablation studies isolating the effects of curriculum learning, hybrid mode, and GRPO modifications
- Discuss the causal relationship between the proposed methods and efficiency improvements

## 3. Unclear Training Procedure for Hybrid Reasoning Modes
**Section 3.2**: The hybrid reasoning mode description lacks critical implementation details. It remains ambiguous whether:
- Both Thinking and NoThinking modes are **explicitly trained** during RL, or
- Only Thinking mode is trained while NoThinking is achieved solely through prompting (e.g., "<thinking> Okay, I think I can solve it directly. </thinking>")

This ambiguity undermines the theoretical analysis (lines 195-206) about mutual benefits between modes. Claims like "a model trained solely in the NoThinking mode may compensate..." and "when trained jointly with the Thinking mode, the model can leverage a compression effect..." assume joint training, but if NoThinking is merely prompt-based inference, these hypotheses lack empirical foundation.

## 4. Inconsistent Explanation of Response Length Trends
**Section 4.3 & Figure 4**: The paper observes opposite trends in response length across training stages - NoThinking responses **increase** while Thinking responses **decrease** - but provides no mechanistic explanation. This observation appears to contradict the hypothesis in Section 3.2 that "NoThinking mode promotes conciseness by reducing verbosity." While the paper acknowledges "NoThinking does not simply mean 'short'", readers need to understand:
- Why NoThinking responses grow longer during training (is this the "compensation effect" mentioned in 3.2?)
- Why Thinking responses become more concise (learned efficiency? curriculum effect?)
- What training dynamics or architectural properties drive this divergence

Without this analysis, it's unclear whether the observed phenomena align with the paper's design intentions or represent unexpected emergent behaviors.

I am happy to raise my score if your response is reasonable.

**Questions:**

## Q1: Clarification on Pure-NoThinking Model in Table 3
**Section 4.3, Table 3**: The paper states "We compare TACLer with the pure NoThinking model trained in the first stage." However, based on the methodology description (Sections 3.1 and 3.3), the first stage only differs in data difficulty (easier problems via curriculum learning), not in the reasoning mode. Could you clarify:
- How is the "Pure-NoThinking" model produced in the first stage?

This clarification is essential to understand the experimental design and validate the claims about hybrid mode benefits.

## Q2: Mechanism Behind Efficiency Improvements
**Section 4.2 & 3.3**: Given that:
1. Your RL objective (GRPO + two tricks) doesn't include explicit length penalties
2. GRPO alone doesn't solve overthinking (as noted in Section 2)
3. Curriculum learning primarily addresses data difficulty, not response verbosity

Could you explain the specific mechanism(s) responsible for the 42.7% reduction in response length? Possible hypotheses to address:
- Do the two GRPO modifications (removing KL, increasing ϵ_high) implicitly encourage conciseness? If so, how?
- Does curriculum learning lead to more efficient reasoning patterns? Why and What evidence supports this?

Clarifying this would strengthen the paper's contribution and help readers understand which aspects of your framework are essential for efficiency.

## Q3: Theoretical Justification for Opposite Length Trends
**Section 4.3, Figure 4(b)**: The observation that NoThinking responses grow longer while Thinking responses grow shorter is intriguing but unexplained. Could you:
- Provide theoretical analysis or intuition for why these opposite trends emerge?
- Verify whether this aligns with your hypothesis in Section 3.2 about "compensation" in NoThinking mode?

Understanding this phenomenon would help assess whether your framework achieves the intended balance between accuracy and efficiency or whether the behaviors emerge for other reasons.

I am happy to raise my score if your response is reasonable.

---

> ### Author Response · Authors · 2025-11-20
> **Response to Reviewer hVt8**
>
> We appreciate that the reviewer found our contributions meaningful in both methodology and practical impact. Thank you also so much for the insightful comments and questions. Below we address each weakness and question, merging them where appropriate.
>
> **Weaknesses 1: minor writing issue**
> We will correct this minor repetition.
>
> **Weaknesses 2 & 4 and Question 2 & 3: Efficiency improvements and trends, and the corresponding underlying mechanisms**
> We appreciate the reviewer’s request for clarification on the learning and reasoning efficiency aspects, which are the core of our work, addressed by our tailored curriculum learning and hybrid reasoning mode, respectively. Below, we address your concerns and questions in detail.
>
> - All RL algorithms used in this paper simply follow previous works as this is not the focus of this work. RL designed for concise reasoning is undoubtedly an interesting direction; for example, as pointed out by reviewers, do the two GRPO modifications (removing KL, increasing ϵ_high) implicitly encourage conciseness? This could be explored in future work.
>
> - The mechanisms of shorter responses. Initially, responses in NoThinking mode are very short, while those in Thinking mode are quite long. Although the two modes varied differently during training, responses in NoThinking mode are still shorter than those in Thinking mode after training. We agree that the analysis section lacks sufficient discussion, which will be addressed in the revision, along the line that we discuss right below.
>
> -  The trends for different modes. As we stated in the methodology section (lines 190-197, which may be somewhat ambiguous, as you pointed out in weakness 3): (i) for NoThinking mode, the model compensates for missing explicit thinking by expanding its reasoning steps, thus leading to a gradual increase in the trend; (ii) for Thinking mode, the model can leverage a compression effect to refine reasoning into shorter responses. We include both modes in our training, hence the trend in the responses. This is evidenced by the generated responses at different stages presented in the appendix. Thanks for raising this point. We will include verification analysis using the model responses in the final revision.
>
> - Regarding curriculum learning, it is indeed primarily used to address data difficulty, not response verbosity, though we show that it can yield more efficient and accurate reasoning in Table 4. Importantly, we report the computation cost and learning efficiency in Table 5 and Figure 5, showing our method cuts substantial training compute. For the mechanism, in the Methodology Section, we indicate that our tailored curriculum learning can determine what knowledge the model lacks and needs to learn in progressive stages, which involves a combined review and learning paradigm. We will clarify that the curriculum learning is not directly targeted at verbosity reduction, rather at a more efficient learning strategy.
>
> - Ablation studies isolating the effects of curriculum learning, hybrid mode. We do have the ablation study on curriculum learning and hybrid mode in the submission (lines 298-322).
>
> **Weaknesses 3: Unclear training procedure for hybrid reasoning modes**
> Thank you for pointing out the ambiguity. We strongly agree with your point that if NoThinking is merely prompt-based inference, the theoretical hypotheses will lack empirical foundation. We are therefore happy to confirm that in our experiments both Thinking and NoThinking modes are explicitly trained at all three stages using RL. We will explicitly state this setup in the revision to avoid any confusion.
>
> **Question 1: Clarification on Pure-NoThinking model in Table 3**
> TACLer is explicitly trained in both Thinking and NoThinking modes across all three stages. For the ablation study, due to computational constraints, we train a model at the first stage using only the NoThinking mode and compare it with TACLer trained at the same stage. We thank the reviewer for raising this point and will clearly articulate this aspect of our methodology and experimental design in the revision.
>
> We agree that addressing the questions raised would strengthen the paper and improve readers' understanding and we’re very grateful to the reviewer for pointing out such aspects. In the revision, we will build upon the responses above by updating: (i) a clear description of the methodology and experimental design; (ii) the causal relationship between the proposed methods and their observed improvements; (iii) the efficiency trend in different modes and the mechanism behind; (iv) hypothesis verification with model responses; and (v) minor issues and other unclear details.

---

### Official Review · Reviewer_3jTy · 2025-10-31

**Soundness:** 3
**Presentation:** 3
**Contribution:** 2
**Rating:** 4
**Confidence:** 3

**Summary:**

This paper proposes TACLer, a reinforcement learning framework that trains models to reason efficiently without losing accuracy. It builds a tailored curriculum that adapts to what the model actually knows, instead of feeding random easy-to-hard data, and trains it progressively through RL. The model also learns to switch between Thinking (long reasoning) and NoThinking (concise reasoning) modes, allowing efficient inference without overthinking.

**Strengths:**

The idea of tailoring the curriculum to the model’s own proficiency is a clean and original way to make RL-based reasoning training more efficient. The hybrid Thinking/NoThinking setup is also elegant. The experiments are well-executed and comprehensive, showing consistent gains across multiple reasoning benchmarks. Overall, it’s a thoughtful and well-structured paper that pushes the discussion on efficient reasoning.

**Weaknesses:**

There seems to be multiple related works that aren't referenced, could the authors comment on the difference to these listed works?

https://arxiv.org/pdf/2505.14970
https://arxiv.org/pdf/2506.06632

**Questions:**

Could the authors do a more recent related work comparison.

---

> ### Author Response · Authors · 2025-11-20
> **Response to Reviewer 3jTy**
>
> We appreciate the recognition of our contributions to tailoring the curriculum and for seeing our hybrid Thinking/NoThinking reasoning modes as elegant, on top of efficient - thank you!
>
> Regarding related works: LLM reasoning is a rapidly evolving field, and many ideas are still in early, exploratory stages, leading to several interesting works appearing at similar times. The two papers you mention are currently available only on arXiv (uploaded 3-5 months before the ICLR deadline), and both are also being reviewed at ICLR 2026.  Indeed, these works explore related ideas on curriculum learning for reasoning. We are happy to engage in discussing how our approach provides a distinct and complementary contribution to them:
>
> - **Key aspects of contemporary works.** (i) Paper [1] dynamically constructs curriculum learning based on curriculum rewards, where the focus is on how reward signals shape the selection of data. The general idea is very close to our tailored curriculum learning, where the paper does that in an online fashion, while ours is offline. (2) Paper [2] introduces a probabilistic scheduler that gradually shifts attention from easy to hard tasks, with the main contribution lying in the scheduling strategy for task difficulty during training. Instead of looking at a probabilistic scheduling strategy, we focus on constructing a tailored curriculum by explicitly leveraging the model’s iterative improvements.
>
> - **Key aspects of our work.** Our contribution is along two dimensions: learning efficiency and reasoning efficiency. We leverage the model’s own iterative improvement to increase data difficulty, thereby constructing a tailored curriculum that enhances learning efficiency. Moreover, we study the hybrid reasoning strategy that improves reasoning efficiency in a way that is orthogonal to the curriculum design. We show that the model benefits from the hybrid setup: (i) In Thinking mode, it produces responses that are both shorter and more accurate than those of the base model; (ii) in NoThinking mode, it generates even shorter responses than in Thinking mode, while achieving higher accuracy than the base model.
>
> - **From an empirical standpoint,** direct comparisons are not straightforward. Neither paper [1] nor paper [2] has released model checkpoints, making replication infeasible. For paper [1], we take accuracy numbers directly from the paper, as shown below, **but the authors did not report solution lengths**, which is a key metric for our work as we also focus on efficiency. The authors of the paper [2] partition benchmarks by difficulty without releasing their splits. We have contacted the authors for access to their models and will update comprehensive results in the final version if they become available.
>
> |Models                   | MATH500  | AIME2024  |
> |--------------------------|----------------|----------------|
> | TACLer (1.5B)        |     88.4       |       42.1     |
> | Paper [1] (3B)      |     67.2     |       10.0      |
> | Paper [1] (7B)      |     76.1     |       17.5      |
>
> Once again, we thank the reviewer for acknowledging our work and will include a detailed comparison and discussion with these related works in the revised version.

---

> > ### Comment · Reviewer_3jTy · 2025-11-24
> >
> > Thank you for the clarification - I will keep my score, because I do not see enough novelty from the prior work.

---

> > > ### Author Response · Authors · 2025-11-26
> > > **Response to Reviewer 3jTy**
> > >
> > > Dear Reviewer 3jTy
> > >
> > >
> > > Thank you for the response. However, we respectfully disagree with the assessment that our work lacks novelty, as we clarified in our previous response. Not only are the two mentioned submissions contemporary to ours and also under review at ICLR, but also: our approach is conceptually distinct from theirs. Interestingly, on top of this, we found that our work already effectively addresses several possible limitations indicated by their reviewers.
> > >
> > > Paper [1] on openreview: https://openreview.net/forum?id=lNgSdqKFmU
> > > Paper [2] on openreview: https://openreview.net/forum?id=KJvHnl3kUv
> > >
> > > **Possible Limitations indicated by ICLR reviewers**
> > > - Paper [1]: The method relies on a predefined or automatically inferred categorization of problem difficulty.
> > > - Paper [1]: As foundation models become increasingly capable, the inherent need for a carefully curated curriculum might decrease, potentially limiting the long-term impact of this method.
> > > - Paper [2]: Model-dependent difficulty: “Easy” and “hard” samples are relative to the specific model; what is easy for one model may be hard for another.
> > > - Paper [2]: Lack of synchronization: Without adaptive scheduling, the scheduler and the model may become misaligned—for instance, the scheduler may advance to harder tasks before the model is ready.
> > > - Paper [2]: Limited Scope of Experiments: Evaluations focus on relatively lower-difficulty tasks
> > >
> > >
> > > **Our work directly addresses these issues**
> > > - No predefined difficulty. Difficulty is built from the model’s own iterative improvement, ensuring alignment with what the model actually needs to learn.
> > > - No scheduler misalignment. Progression depends on the model’s demonstrated capability, preventing premature advancement.
> > > - High-difficulty benchmarks. We include more challenging evaluations, such as AIME 2024/2025.
> > >
> > >
> > > **We contribute an additional, orthogonal innovation: hybrid reasoning**
> > > - Neither of the papers mentioned proposes improvements to reasoning efficiency. Our hybrid Thinking/NoThinking strategy is a second, independent contribution.
> > >
> > >
> > > Given these distinctions, we believe our paper provides substantial and complementary contributions beyond the two contemporary submissions.
> > >
> > >
> > > We kindly hope the reviewer finds our clarifications, and we would be grateful if the reviewer could reconsider their evaluation.

---

### Official Review · Reviewer_Xdod · 2025-11-01

**Soundness:** 2
**Presentation:** 3
**Contribution:** 1
**Rating:** 2
**Confidence:** 4

**Summary:**

The authors present a simple curriculum strategy for training for reasoning with GRPO. They propose to assess the model's accuracy on the questions first to go from easy to hard questions, which in combination with the hybrid reasoning idea, seem to improve the model's reasoning further than others. But, there are a few works [1, 2] that explored exactly this idea and am skeptical of the novelty of the idea here despite strong results.

[1] Parashar, Shubham, et al. "Curriculum Reinforcement Learning from Easy to Hard Tasks Improves LLM Reasoning." arXiv preprint arXiv:2506.06632 (2025).
[2] Ji, Yunjie, et al. "How Difficulty-Aware Staged Reinforcement Learning Enhances LLMs' Reasoning Capabilities: A Preliminary Experimental Study." arXiv preprint arXiv:2504.00829 (2025).

**Strengths:**

- The paper is clearly written and the proposed contribution of designing a curriculum based on the current model's success rate is warranted. They also show strong results on the math datasets, and it interesting how the hybrid reasoning mode seems to yield concise reasoning.

**Weaknesses:**

- The main limitation that I see with the paper is novelty. There are mainly two works cited in the summary [1, 2] that seem to propose the same idea and they do not compare against these baselines. If the authors tackle this point effectively, I would be willing to revise my score.

**Questions:**

- What does it mean to have complete solution but wrong final answer? Does that just mean formatting error for the answer?
- How is the hybrid mode actually used? Which questions are trained with the thinking and no-thinking?

---

> ### Author Response · Authors · 2025-11-20
> **Response to Reviewer Xdod**
>
> Thank you for the time and review. We address the raised concerns as follows:
>
> **Comparison with contemporary works and the novelty of our work**
> LLM reasoning is a rapidly evolving field, and many ideas are still in early, exploratory stages with many interesting works appearing at the same time. The two papers you mention are currently available only on arXiv (uploaded 3-5 months before the ICLR deadline), and one of them is under review for ICLR 2026.  Indeed, these works explore related ideas on curriculum learning for reasoning. We appreciate the opportunity to clarify how our approach provides a distinct and complementary contribution.
>
> - **Conceptual Comparison.** Although we share the general curriculum-like ideas, our approach is distinct in both mechanism and objective.
>  (1) Paper [1] introduces a probabilistic scheduler that gradually shifts focus from easy to hard tasks; their contribution lies in the scheduling strategy for training data difficulty.
>  (2) Paper [2] employs multiple models (from smaller to larger ones) to filter data, and proposes a progressively staged training pipeline involving SFT and RL.
>  (3) Our work contributes along two dimensions: learning efficiency and reasoning efficiency. We leverage the model’s own iterative improvement to increase data difficulty, thereby constructing a tailored curriculum that enhances learning efficiency in a manner fundamentally different from the above works. Moreover, we study the hybrid reasoning strategy that improves reasoning efficiency in a way that is orthogonal to the curriculum design. We show that the model benefits from the hybrid setup: (i) In Thinking mode, it produces responses that are both shorter and more accurate than those of the base model;  (ii) in NoThinking mode, it generates even shorter responses than in Thinking mode, while achieving higher accuracy than the base model.
>
> - **Empirical Comparison.** Neither paper has released model checkpoints, making a direct comparison infeasible. The authors of the paper [1] split benchmarks by difficulty but did not release these splits. For paper [2], we report the accuracy numbers directly from their paper as shown below; however, **they do not report solution length**, which is a key metric for our work since we stress efficiency. We have contacted the authors for access to their models and will include more comprehensive comparisons in the updated version if they become available.
>
> |Models                   | MATH500  | AIME2024  |
> |--------------------------|----------------|----------------|
> | TACLer                  |     88.4       |       42.1      |
> | Paper [2]                |     89.5      |       42.3      |
>
>
> **Question 1: What does it mean to have complete solution but wrong final answer? Does that just mean formatting error for the answer?**
> ‘Complete solution but wrong final answer’ refers to cases where the model generates a full reasoning process within the maximum sequence length, but the final answer is incorrect (not merely a formatting error). We can change the phrasing “complete solution” to the sentence we just wrote as explanation here.
>
> **Question 2: How is the hybrid mode actually used? Which questions are trained with the thinking and no-thinking?**
>  During the training, each question is used for both Thinking and NoThinking modes. For NoThinking, we explicitly prompt the model to solve the problem without thinking: \<thinking\> Okay, I think I can solve it directly. \</thinking\>; otherwise, the model operates in Thinking mode. This setting is applied consistently during both training and evaluation.
>
> We will incorporate these clarifications, together with the discussion and comparison of related works, in the revised version.

---

### Author Response · Authors · 2025-12-01
**Quick Summary of Review Issue and Response**

We thank all reviewers for the time and effort invested in reviewing our submission. Below, we summarise the key points raised and our views and response.

**Reviewer Xdod01 and 3jTy** compare the novelty of our work to **three contemporary arXiv papers (two of them are also under review at ICLR 2026)**. Our rebuttal clarifies that although these works explore curriculum-style ideas, our method is mechanistically distinct:

- We leverage the model’s own iterative improvement to increase data difficulty, thereby constructing a tailored curriculum that enhances learning efficiency in a manner **fundamentally different from the related works**;
- We introduce a hybrid reasoning strategy that improves reasoning efficiency in a way that is orthogonal to the curriculum design.
- Moreover, our work already effectively addresses several possible limitations in the referenced works, for example by introducing the model-tailored curriculum.

We also note that: (i) the review of Reviewer 3jTy only consists of two recommended papers under review for ICLR 2026, and does not offer any substantive evaluation of our work, which we believe is not valid criticism; (ii) **conversely, Reviewer Xdod01 explicitly indicates willingness to raise scores upon satisfactory clarification**.

**Reviewer hVt8** acknowledges our paper's meaningful methodological and practical contributions, particularly the tailored curriculum and hybrid reasoning framework. The concerns primarily involve clarification rather than pointing to errors: (i) methodological and experimental design; (ii) the causal relationship between the proposed method and its trends and improvements; (iii) the use of model responses to validate hypotheses. We agree that addressing the questions raised would strengthen the paper, and provide our revised plan in our response. **The reviewer explicitly states that they would be willing to raise their score once the concerns are addressed**.

**Reviewer kYe1** finds the paper interesting and well-written, noting that TACLer’s hybrid reasoning strategy and accuracy-efficiency trade-offs are clearly presented. The reviewer’s concerns centre on experimental scope, cost analysis, and upper bound. Importantly, these concerns relate to clarifications rather than methodological errors. We provide detailed clarifications on the experimental setting and cost in our response. In addition, we follow the reviewer’s insightful suggestion to compute an oracle upper bound, showing TACLer achieves over 86% of this ceiling. Overall, the concerns are addressable clarifications rather than fundamental issues.

We would like to thank all reviewers again and will incorporate the corresponding revisions into our paper.

---

### Meta-Review · Area_Chair_4ByC · 2026-01-07

**Summary:**

The work introduces a simple curriculum strategy for training for reasoning with GRPO. It proposes to assess the model's accuracy on the questions first to go from easy to hard questions, which in combination with the hybrid reasoning idea, seem to improve the model's reasoning further than others. However, there are some works that have also explored similar ideas. Thus the novelty of this work is not significant enough. All reviewers recommend rejection.

**Reviewer Concerns:**

There are some works that have also explored similar ideas. Thus the novelty of this work is not significant enough. This concern is not well addressed.

**Reviewer Scores:**

Reviewers are likely to keep the scores.

---

### Decision · Program_Chairs · 2026-01-26

Reject